# UniGM: Unifying Multiple Pre-trained Graph Models via Adaptive Knowledge Aggregation

## ABSTRACT

Recent years have witnessed remarkable advances in graph representation learning using Graph Neural Networks (GNNs). To fully exploit the unlabeled graphs, researchers pre-train GNNs on large-scale graph databases and then fine-tune these pre-trained Graph Models (GMs) for better performance in downstream tasks. Because different GMs are developed with diverse pre-training tasks or datasets, they can be complementary to each other for a more complete knowledge base. Naturally, a compelling question is emerging: *How can we exploit the diverse knowledge captured by different GMs simultaneously in downstream tasks?* In this paper, we make one of the first attempts to exploit multiple GMs to advance the performance in the downstream tasks. More specifically, for homogeneous GMs that share the same model architecture but are obtained with different pre-training tasks or datasets, we align each layer of these GMs and then aggregate them adaptively on a per-sample basis with a tailored Recurrent Aggregation Policy Network (RAPNet). For heterogeneous GMs with different model architectures, we design an alignment module to align the output of diverse GMs and a meta-learner to decide the importance of each GM conditioned on each sample automatically before aggregating the GMs. Extensive experiments in various downstream tasks from 3 domains reveal our dominance over each single GM. Additionally, our methods (UniGM) can achieve better performance with moderate computational overhead compared to alternative approaches including ensemble and model fusion. Also, we verify that our methods are not limited to graph data but could be flexibly applied to multiple modalities. The codes can be seen in the anonymous link: https://anonymous.4open.science/r/UniGM-DA65.

## CCS CONCEPTS

• **Computing methodologies → Neural networks**; **Learning latent representations**.

## KEYWORDS

Graph analysis, pre-trained models, ensemble, model fusion

## 1 INTRODUCTION

Fine-tuning a pre-trained Language Model (LM) has become the de facto standard for Natural Language Processing (NLP) [4, 6]. Inspired by the prosperity, tremendous efforts have been devoted

*ACM MM, 2024, Melbourne, Australia*
© 2024 Copyright held by the owner/author(s). Publication rights licensed to ACM.
ACM ISBN 978-x-xxxx-xxxx-x/YY/MM
https://doi.org/10.1145/nnnnnnn.nnnnnnn

to pre-trained GMs to exploit abundant knowledge of unlabelled graphs [18, 53]. For the pre-training stage, researchers train the GNN encoder with various pretext tasks [35]. For the fine-tuning stage, researchers replace the top layer of the pre-trained models with a task-specific sub-network and train the new model with the labeled data of the downstream tasks. Pre-training techniques can help GNNs capture the potential laws of graph data that are conducive to downstream tasks [18, 53]. Intuitively, different off-the-shelf GMs are obtained with diverse pre-training tasks or datasets and thus they capture diverse knowledge and possess different abilities. Take molecular graphs as examples, given that motifs in molecular graphs usually correspond to functional groups that are indicative of molecular properties, some researchers pre-train GNNs with motif-driven pre-training strategy [61] to capture the information of functional groups. Now, we are naturally motivated to ask the following question: *How can we exploit the diverse knowledge captured by different GMs simultaneously in downstream tasks?*

There are several possible approaches to achieving this goal. For example, the easiest way is to adopt all the pretext tasks to pre-train only one model on various datasets. However, it is impractical because the downstream users are often only accessible to the off-the-shelf pre-trained GMs rather than the pre-training datasets or tasks. Worse still, pre-training a new model from scratch with multiple tasks and datasets is computationally prohibitive. Therefore, we consider unifying the off-the-shelf pre-trained GMs during model adaptation. Ensemble Learning [9] is a prevalent technique that can unify multiple models. Despite the effectiveness, we have to fine-tune each GM and then use the averaged outputs of them for downstream tasks, which is inconvenient and suffers from heavy computational overhead. Model fusion [1, 32, 38] is another alternative solution to this problem, which aligns neurons across different models before averaging their associated parameters in a data-free way. While model fusion enjoys higher efficiency than ensemble learning, there is a flaw that causes poorer performance: it treats all the samples equally by letting them share the same aggregation policy. However, in practice, each sample holds specific relations with diverse pre-trained models [56] and the aggregation policy should depend on each sample. Additionally, existing evidence reveals that the lower pre-trained layers learn more general features while the higher layers closer to the output specialize more to the pre-training tasks [20, 60]. Therefore, for some downstream tasks that are more similar to pre-training tasks, the aggregation should emphasize the higher layer and vice versa. Overall, an ideal aggregation policy should be both sample-dependent and layer-dependent. Also, tremendous efforts have been devoted to designing pre-training strategies for GNNs so far. However, how to leverage pre-trained GNNs remains under-explored.

To remedy the above drawbacks, we propose UniGM to exploit multiple GMs effectively and efficiently during fine-tuning.

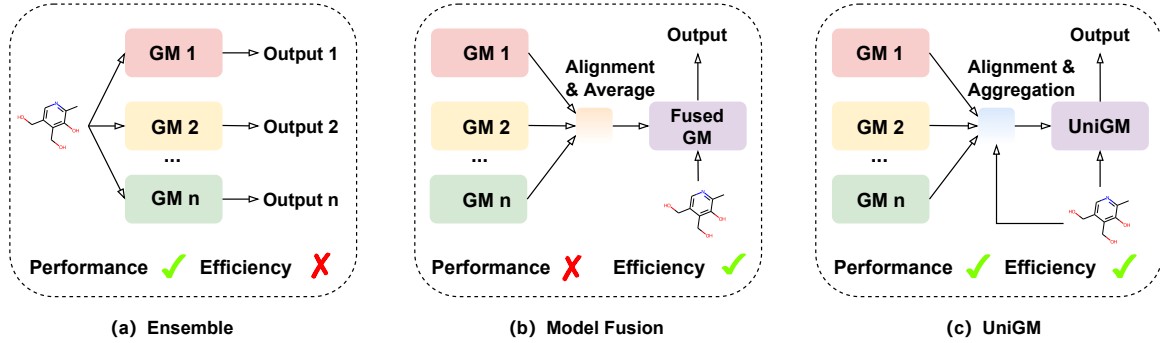

**Figure 1: Comparison of the ensemble, model fusion, and UniGM.**

We show the schematic diagrams of our UniGM and the above-mentioned approaches in Figure 1. Specifically, for homogeneous GMs that share the same GNN backbone, we aggregate each layer of them adaptively on a per-sample basis with a tailored RAPNet, which includes a Recurrent Neural Network (RNN) [31] to explicitly model the layer-based relations. For the heterogeneous GMs with different GNNs backbones, we devise an alignment module to align the output of heterogeneous GMs and a meta-leaner to decide the importance of each GM for the downstream task conditioned on per sample automatically. Here, 'Heterogeneous GMs' denote the pre-trained graph models that differ from each other in terms of GNN backbones, instead of heterogeneous graph data or heterogeneous GNNs [59]. Different from some recent works that aim to combine several self-supervised tasks to pre-train GNNs [13, 22], we attempt to unify multiple off-the-shelf pre-trained GMs for a more complete knowledge base. We highlight the following contributions:

- Currently, the community focuses on designing self-supervised pre-training strategies for GNNs, however, it remains under-explored how to utilize pre-trained GMs more effectively or efficiently. To the best of our knowledge, we make one of the first attempts to unify multiple GMs for better performance in downstream tasks.
- We present two effective and efficient techniques to unify homogeneous and heterogeneous GMs, respectively. Our methods can also be flexibly applied to various modalities (validated in section 4.5).
- Extensive experiments validate that UniGM can consistently outperform each single GM, and achieve state-of-the-art performance with moderate computational consumption compared with competitive alternatives.

## 2 RELATED WORK

### 2.1 Pre-training Graph Neural Networks

GNNs have emerged as dominant tools for graph representation learning. While effective and prevalent, they require expensive annotations and barely generalize to unseen graphs, which poses a hurdle to practical applications. To remedy these deficiencies, tremendous efforts have been devoted to pre-training GNNs. One line of these works follows the contrastive paradigm [14, 34, 44, 62]. For

example, GraphCL [58] and its variants [11, 26, 42, 43, 48, 51, 57] embed augmented versions of the anchor graph close to each other and push the embeddings of other graphs apart. Additionally, DGI [46] and InfoGraph [41] is proposed to obtain expressive representations for graphs or nodes via maximizing the mutual information between graph-level representations and substructure-level representations of different granularity. The other line of work adopts generative or predictive pretext tasks. Typically, GPT-GNN [19] introduces an attributed graph generation task to pre-train GNNs so that they can capture the structural and semantic properties of the graph. Additionally, [18], [25] and [17] conduct attribute and structure prediction at the level of individual nodes as well as entire graphs. To capture the rich information in molecular graph motifs, GROVER [35] and MGSSL [61] propose to predict or generate the motifs. Considering that 3D geometric information also plays a vital role in predicting molecular graph properties, several recent works [10, 27, 28, 40] pre-train the GNN encoders on molecular datasets with 3D geometric information. Since the above GMs are obtained with diverse pre-training tasks or datasets, they can be complementary to each other. To this end, we propose UniGM to integrate multiple GMs into a unified one for better performance.

### 2.2 Ensemble Learning and Model Fusion

Ensemble Learning has achieved spectacular achievements in history [37, 49]. They combine the outputs of different models to improve performance. In the pretrain-then-finetune paradigm, we have to finetune all the pre-trained models and then run each of them during inference to average their outputs, which is laborious. Alternatively, Model Fusion aims to merge multiple trained networks into a single one in a data-free manner. The simplest way of model fusion is vanilla averaging the parameters of pre-trained networks [45]. However, vanilla averaging only works in the case when the weights of individual networks are relatively close in the weight space. As effective remedies, FBA-Wagging [1], FedMA [47] and OTFusion [38] align the neurons of each layer before applying vanilla averaging. Although model fusion runs several magnitudes faster than ensemble learning, the fusion process is independent of the input sample while each sample holds specific relations with diverse models, which accounts for its poorer performance. Compared with them, our UniGM achieves better performance with moderate computational cost.

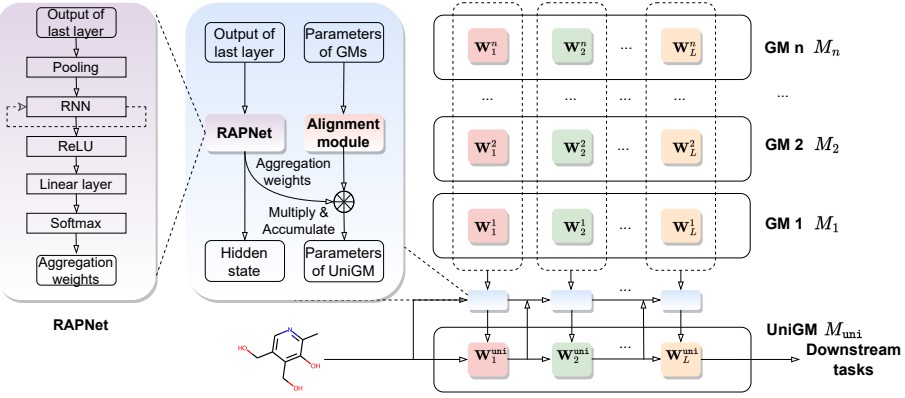

**Figure 2: Schematic diagram of UniGM for homogeneous GMs.**

## 3 METHOD

Our UniGM encompasses two ingredients: Unifying homogeneous GMs and Unifying heterogeneous GMs. In what follows, we elaborate on them in detail.

### 3.1 Unifying homogeneous GMs (UniGM)

As shown in Figure 2, given $n$ homogeneous GMs $\mathcal{M} = \{M_1, M_2, ..., M_n\}$ with the same backbone, we aggregate their parameter matrices layer-wisely following the 'Alignment-then-Aggregation' paradigm. We consider that $i$-th GM $M_i$ consists of $L$ layers whose parameter matrices are $\mathbf{W}_1^i, \mathbf{W}_2^i, ..., \mathbf{W}_L^i$. Next, taking $j$-th layer as an example, we elaborate on the alignment and aggregation modules to obtain the $j$-th layer parameter matrix $\mathbf{W}_j^{uni}$ of the unified model $M_{uni}$.

**Alignment Module.** Since the homogeneous GMs are pre-trained with different tasks or datasets, so even the parameters at the same layer of them may contain different semantic meanings, which hinders direct aggregation. To tackle this issue, we can feed the parameter matrices to linear layers to project them to a shared weight space to align them. However, this way will incur heavy computation with multiple matrix multiplications. Hence, we use lighter convolution. Specifically, given $n$ parameters matrices $\mathbf{W}_j^{\{1,2,\cdots,n\}}$ for $j$-th layer, each of which are of scale $H_{in} \times H_{out}$, we resize them as a $1 \times n \times H_{in} \times H_{out}$ tensor $\overline{\mathbf{W}}_j$ and feed it to a pointwise convolution layer including $n$ filters $\mathbf{C}_j^{\{1,2,\cdots,n\}}$, each of which is with kernel size $n \times 1 \times 1$. The output $\widehat{\mathbf{W}}_j$ of size $n \times H_{in} \times H_{out}$ are regarded as the aligned parameter matrices. The process can be formulated as $\widehat{\mathbf{W}}_j^i = \mathbf{C}_j^i * \overline{\mathbf{W}}_j$, where '$*$' is the convolution with time complexity $O(n^2 H_{in} H_{out})$. It is superior to the linear layer of size $H_{out} \times H_{out}$ with complexity $O(n H_{in} H_{out}^2)$ because $H_{out} \gg n$ in practice. Kindly note that we initialize the convolution as an identical mapping for a warm-up from pre-trained parameters.

**Aggregation Module.** As we discuss in the introduction section, the aggregation policy should be both sample-dependent and layer-dependent. To this end, we introduce a Recurrent Aggregation Policy Network (RAPNet) which is conditioned on the input feature of each layer to learn the aggregation policy for the aligned parameter matrices. The term "aggregation policy" refers to the

weights used to linearly combine the aligned parameter matrices into unified ones. Specifically, for each layer, we first apply a global pooling to transform the input feature into a one-dimensional embedding vector, which will be fed into the RNN [31] to model the dependencies between different layers. Namely, we regard the one-dimensional embedding vector of each layer as the input for a timestamp in RNN and the hidden state of RNN will be propagated to the next layer. Formally, for $j$-th layer, given that the input feature (after pooling) is $\widehat{h}_j$, we can obtain the output of RNN $o_j$ by,

$$s_j = tanh(\mathbf{P}\widehat{h}_j + \mathbf{Q}s_{j-1} + b), o_j = tanh(\mathbf{R}s_j + c), \quad (1)$$

where $s_j$ is the hidden state of layer $j$ and we initialize $s_0$ with zeros. $\mathbf{P}, \mathbf{Q}, \mathbf{R}$ are the parameters of the RNN. $b$ and $c$ are the bias terms. Finally, we transform the output of the RNN ($o_j$) to the aggregation weights (policy) with a fully-connected layer followed by a softmax function, i.e., $A_j(h_{j-1}) = \text{Softmax}(\text{Linear}(\text{ReLU}(o_j)))$. The $i$-th dimension of $A_j(h_{j-1})$ is $A_j^i(h_{j-1})$, which denotes the learned aggregation weights (policy) for the $j$-layer parameter matrix of the $i$-th pre-trained model ($\widehat{\mathbf{W}}_j^i$). Finally, we can obtain the $j$-th layer parameter matrix $W_j^{uni}$ of the unified model by re-weighting the aligned matrices with the learned aggregation policy,

$$\mathbf{W}_j^{uni} = \sum_{i=1}^{n} A_j^i(h_{j-1})\widehat{\mathbf{W}}_j^i. \quad (2)$$

With the aggregated parameters in the unified model $M_{uni}(\cdot; \mathbf{W}_1^{uni}, \mathbf{W}_2^{uni}, \cdots, \mathbf{W}_L^{uni})$, we can formulate the loss as,

$$\mathcal{L} = \mathbb{E}_{(\mathbf{x},\mathbf{y}) \sim \mathcal{D}} \ell\left(M_{uni}\left(\mathbf{x}; \mathbf{W}_1^{uni}, \mathbf{W}_2^{uni}, \cdots, \mathbf{W}_L^{uni}\right), \mathbf{y}\right), \quad (3)$$

where $\mathcal{D} = \{(\mathbf{x}, \mathbf{y})\}$ denotes the dataset of downstream tasks and $\mathbf{x}, \mathbf{y}$ denote the sample and label. $\ell$ is the loss of downstream tasks. We provide two variations for UniGM. The first one, dubbed UniGM-F, is to freeze the pre-trained parameters of GMs and only tune the parameters of alignment and aggregation modules. The other one named UniGM-T is to tune all the parameters. Unlike ensemble learning, UniGM is more efficient because the samples are only required to pass through the unified model while the samples in ensemble learning need to pass through all the GMs. Compared with

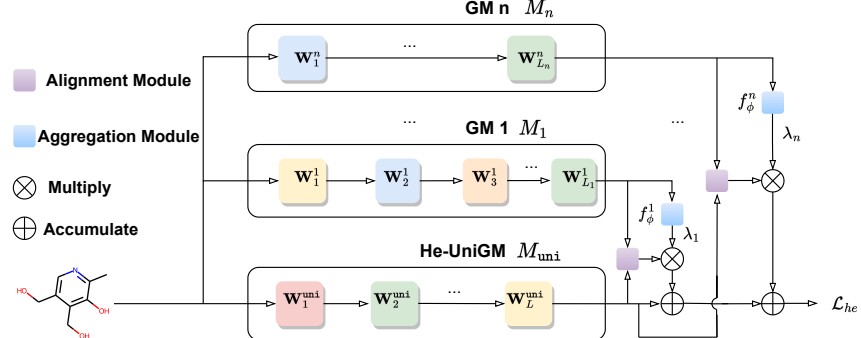

**Figure 3: The schematic diagram for unifying heterogeneous GMs (He-UniGM).**

model fusion, UniGM aggregates parameters of GMs adaptively depending on the sample and layer, leading to better performance.

## 3.2 Unifying heterogeneous GMs (He-UniGM)

Although most current open-sourced GMs for the same domain share the same GNN encoder, future GMs may adopt more powerful GNNs. However, UniGM-T and UniGM-F cannot unify heterogeneous GMs. As a remedy, we develop another effective strategy (He-UniGM) to integrate heterogeneous GMs into a unified one, whose general pipeline can be seen in Figure 3.

**Alignment Module.** Since heterogeneous GMs are separately pre-trained with different networks or datasets, both the semantics and dimensions of their outputs are not well-aligned. We introduce the following strategy to overcome this issue. Specifically, let $M_i(\cdot), M_{uni}(\cdot)$ be the output of $i$-th GM and He-UniGM respectively, we minimize the following $\ell_2$ objective to align their feature space,

$$\|R_\omega\left(M_{uni}(\mathbf{x}; \theta_{uni})\right) - M_i(\mathbf{x}; \theta_i)\|_2^2, \tag{4}$$

where $R_\omega(\cdot)$ is a linear transformation parameterized by $\omega$. Different from homogeneous settings, the parameters $\theta_{uni}$ of the unified model are initialized randomly and updated with the following aggregation module.

**Aggregation Module.** Considering that diverse GMs contribute unequally to the downstream task, we introduce a learnable parameter $\lambda_i$ to automatically decide the importance of GM $M_i$. We set $\lambda_i = f_\phi^i(M_i(\mathbf{x}; \theta_i))$ in order to model the importance of $M_i$ conditioned on the input $\mathbf{x}$, where $f_\phi^i(\cdot)$ is a light meta-learner (1-layer fully-connected network in practice) parameterized by $\phi$. We can then formulate the loss of aggregation as,

$$\mathcal{L}_{agg} = \mathbb{E}_{(\mathbf{x},\mathbf{y}) \sim \mathcal{D}} \sum_{i=1}^{n} \lambda_i \|R_\omega\left(M_{uni}(\mathbf{x}; \theta_{uni})\right) - M_i(\mathbf{x}; \theta_i)\|_2^2, \tag{5}$$

where $n$ is the number of GMs. And then, the optimization objective of He-UniGM is,

$$\mathcal{L}_{he} = \mathcal{L}_{task} + \alpha * \mathcal{L}_{agg}, \tag{6}$$

where $\alpha$ is a hyper-parameter and $\mathcal{L}_{task}$ is the loss of downstream task,

$$\mathcal{L}_{task} = \mathbb{E}_{(\mathbf{x},\mathbf{y}) \sim \mathcal{D}} \ell\left(M_{uni}\left(\mathbf{x}; \theta_{uni}\right), \mathbf{y}\right). \tag{7}$$

Then, we utilize $\varphi$ to denote both the parameters of linear transformation $\omega$ and unified model $\theta_{uni}$ for convenience. We can solve above problem with following bilevel scheme [2, 12, 21],

$$\min_\phi \quad \mathcal{L}_{task}\left(\varphi^*\right), \text{s.t.} \quad \varphi^* = \operatorname{argmin}_\varphi \mathcal{L}_{he}(\varphi, \phi). \tag{8}$$

In practice, we can choose gradient descent (GD) to approximately solve the inner optimization,

$$\varphi_{t+1} = \varphi_t - \beta \nabla_\varphi \mathcal{L}_{he}\left(\varphi_t, \phi\right), \tag{9}$$

where $\beta$ is the learning rate. Now we consider solving the outer optimization with gradient-based methods. The prerequisite is the gradients of $\mathcal{L}_{task}$ w.r.t $\phi$. Let $\varphi_T$ is the approximate optimal solution obtained with $T$ steps GD in Eq.(9), we can then re-write the gradients as,

$$\nabla_\phi \mathcal{L}_{task}(\varphi_T) = \nabla_\varphi \mathcal{L}_{task}(\varphi_T)\nabla_\phi\varphi_T, \tag{10}$$

where the gradient $\nabla_\phi\varphi_T$ can be computed by unrolling the dynamics of the inner loop from $\varphi_T$ to $\varphi_0$. In the forward computation, successive parameters $\varphi_0, \cdots, \varphi_T$ are cached. In the backward call, the cached parameters are used to compute gradients in a series of vector-jacobian products. During the reverse computation, the gradient starting from the $\nabla_\phi\varphi_T$ can be propagated to the intermediate parameters $\varphi_t$ through $\nabla_{\varphi_t}\varphi_{t+1}$:

$$\nabla_{\varphi_t}\varphi_{t+1} = 1 - \beta\nabla_{\varphi_t}^2 \mathcal{L}_{he}\left(\varphi_t\right), \quad t \in \{0, \ldots, T-1\}, \tag{11}$$

where $\nabla_{\varphi_t}^2$ is the Hessian. We can then obtain the gradients $\mathcal{L}_{task}$ w.r.t $\phi$ with,

$$\nabla_\phi\mathcal{L}_{task}(\varphi_T) = \nabla_\varphi\mathcal{L}_{task}(\varphi_T) \sum_{t=T-1}^{0} \left[\nabla_{\varphi_{t+1}}\varphi_T\right]\nabla_\phi\varphi_{t+1}$$

$$= -\beta\nabla_\varphi\mathcal{L}_{task}(\varphi_T) \sum_{t=T-1}^{0} \left[\nabla_{\varphi_{t+1}}\varphi_T\right]\nabla_\phi\left(\nabla_{\varphi_t}\mathcal{L}_{he}\left(\varphi_t, \phi\right)\right), \tag{12}$$

where $\nabla_{\varphi_{t+1}}\varphi_T$ can be iteratively derived with Eq. (11). Kindly note that the bilevel optimization can be done efficiently with PyTorch [33] because (1) $\varphi$ only includes the parameters of the linear transformation and the unified model; (2) $T = 2$ is enough in our experiments. Compared with the ensemble, He-UniGM is computationally cheaper because (1) the parameters of multiple GMs are

frozen during the training stage; (2) He-UniGM only uses the unified model (one model) for inference.

## 4 EXPERIMENTS

### 4.1 Experimental Settings and Baselines.

Following previous works on the topic of pre-training GNNs [18, 30], we evaluate UniGM on 3 downstream tasks from 3 domains: molecular property prediction in chemistry, protein function prediction in biology, and research field prediction in the bibliography.

For the first task, we adopt the 8 binary classification datasets contained in MoleculeNet [50]. For the second task, we use protein-protein interaction (PPI) networks consisting of 88K proteins from 8 different species, where the subgraphs centered at a protein of interest (i.e., ego-networks) are used to predict their biological functions. The task is to predict 40 fine-grained biological functions corresponding to 40 binary classification tasks. For the third task, we predict the research field with 299,447 labeled subgraphs from 6 different categories. We randomly split the downstream data and evaluate test performance with micro-averaged F1 score. Additionally, we evaluate UniGM on more downstream tasks in the experiments. For homogeneous UniGM, we unify recent open-sourced GMs including GraphCL, MGSSL, SimGRACE, and GraphMVP in chemistry and Infomax, EdgePred, ContextPred, AttrMask for both the biology and bibliography domains. For heterogeneous GMs in chemistry, we first pre-train different GNNs with the pre-training tasks proposed in the above works. And then, we integrate the obtained GMs into a unified one with He-UniGM. For single GM, we report the results of baselines in Table 1. Additionally, we consider several alternatives that can also utilize multiple GMs. Specifically, 'Vanilla Average' refers to we use the average of the weights of GMs to initialize a new model for prediction. 'Concatenation' denotes the baselines that we take the graph embeddings from the pre-trained models, concatenate them, and pass them into a single linear layer to finetune w.r.t the downstream task.

For model fusion, we adopt the most advanced method OTFusion [38] so far. For homogeneous GMs, we set the learning rate as $1 \times 10^{-3}$. The hidden size of RNN in RAPNet is set as 8 and the number of layers is 2. *Note that we only aggregate the fully-connected layers of GNNs. The embedding layers and the batch normalization layers of each GM are not integrated into a unified one.* For heterogeneous settings, we unify heterogeneous GMs with diverse GNN architectures. We provide the details of these heterogeneous GMs in the appendix. For the chemistry and biology domains, we adopt a 5-layer Graph Isomorphism Networks (GINs) [54] whose hidden dimension is 300 as the backbone architecture, which is one of the most expressive GNNs. In the fine-tuning stage, we use a batch size of 32 and dropout rate of 50%. On the molecular property prediction datasets, we train models for 100 epochs, while on the protein function prediction dataset (with the 40 binary prediction tasks), we train models for 50 epochs. All the above models are trained with Adam optimizer with a learning rate of 0.001 and we evaluate test performance on downstream tasks using ROC-AUC. For bibliography domain, we train the pre-trained GNNs with Adam optimizer with a learning rate of 0.001 and batch size as 32 for 50 epochs. In all the 3 domains, the split for train/validation/test sets is 80% : 10% : 10%. We use ADAM optimizer for training the meta-networks

with a learning rate of $1 \times 10^{-3}$. Additionally, we set the steps of inner optimization as 2 (i.e., $T = 2$). Hyper-parameter $\alpha$ is picked from $\{0.1, 0.2, 0.5, 0.8\}$ with the validation set. All experiments are conducted on Tesla V100 GPUs. *More details can be found in the appendix.*

### 4.2 Results and Analysis.

Table 1, Table 2, and Table 3 document the main results in terms of accuracy. Table 4 and Table 5 compare the computational efficiency, from which we make the following observations (Obs):

Obs 1. Variants of UniGM achieve notable improvements over every single model. However, they inevitably introduce extra computational costs. *We compare the memory consumption in the appendix.*

Obs 2. Variants of UniGM achieve better performance while enjoying higher efficiency than ensemble in most cases. Although model fusion is more efficient than UniGM, its performance is unsatisfactory and even sometimes inferior to the single model. Moreover, model fusion cannot work in heterogeneous settings. Overall, UniGM achieves better performance with moderate computational budgets.

Obs 3. UniGM-F performs better than UniGM-T in datasets with smaller scales while the latter is superior in larger-scale datasets. This phenomenon coincides with the observations of a recent work [52]: the over-parameterized models tend to overfit the limited labeled graphs. UniGM-T with more learnable parameters is more likely to overfit the small-scale datasets. To support these claims, we plot the training and testing accuracy curves in the appendix.

### 4.3 Case Study

In this section, we study whether UniGM can possess the specialized abilities of the GMs it is composed of. We adopt two tasks: 3D Diameter Prediction [28] and Atom Type Prediction [18]. The former means using 2D molecular graph to predict the 3D diameter, which is challenging with respect to the 2D topology but straightforward using 3D geometry because the 2D and 3D landscapes of some molecules are considerably different (Figure 4). The latter means predicting atoms' type. As shown in Figure 4, GraphMVP [28] performs the best in 3D Diameter Prediction because it can capture the 3D geometry. Analogously, AttrMask [18] is better at Atom Type Prediction. UniGM composed of GraphMVP and AttrMask possesses their unique abilities, which verify that UniGM constitutes a more complete knowledge base.

### 4.4 Ablation Study

**GMs' diversity.** Although UniGM achieves impressive results, it remains to be explored: What the performance gains can be attributed to? The GMs' diversity or more learnable parameters? In Table 6, we substitute diverse GMs in UniGM-T and UniGM-F with the same one and keep the number of GMs unchanged. '4 × MGSSL' means that we substitute 4 GMs in UniGM-T or UniGM-F with 4 MGSSL models. In this way, we keep the number of learnable parameters unchanged while observing the role of GMs' diversity. We can draw the following conclusions: (1) More parameters are not necessarily conducive for downstream tasks. Since most datasets in experiments are insufficiently labeled, over-parameterized models

**Table 1: Results for molecular property prediction tasks (homogeneous setting). We report the mean (and standard deviation) ROC-AUC of 10 seeds with scaffold splitting. The best results and the second best are highlighted with bold and bold, respectively. We also highlight the performance of the GMs that UniGM contains with the gray background. 'No pretrain' means training from scratch. The original papers marked with '◇' did not follow the standard fine-tuning settings, which we elaborate on in the appendix. For fairness, we reproduce their fine-tuning results following the settings of the pioneering work [18]. Considering that the std is relatively large on small-scale molecular datasets, we highlight the results that outperform the best baselines with ≥ 0.5 std / ≥ 2 std with '★' and '+' respectively to show how statistically significant the improvement is.**

| | Tox21 | ToxCast | Sider | ClinTox | MUV | HIV | BBBP | Bace | Average |
|---|---|---|---|---|---|---|---|---|---|
| # graphs | 7,831 | 8,575 | 1,427 | 1,478 | 93,087 | 41,127 | 2,039 | 1,513 | - |
| No pretrain | 74.6 (0.4) | 61.7 (0.5) | 58.2 (1.7) | 58.4 (6.4) | 70.7 (1.8) | 75.5 (0.8) | 65.7 (3.3) | 72.4 (3.8) | 67.15 |
| InfoGraph [41] | 73.3 (0.6) | 61.8 (0.4) | 58.7 (0.6) | 75.4 (4.3) | 74.4 (1.8) | 74.2 (0.9) | 68.7 (0.6) | 74.3 (2.6) | 70.10 |
| EdgePred [18] | 76.0 (0.6) | 62.8 (0.6) | 60.4 (0.7) | 64.1 (3.7) | 75.1 (1.2) | 76.3 (1.0) | 67.3 (2.4) | 77.3 (3.5) | 70.08 |
| AttrMasking [18] | 75.1 (0.9) | 63.3 (0.6) | 60.5 (0.9) | 73.5 (4.3) | 75.8 (1.0) | 75.3 (1.5) | 65.2 (1.4) | 77.8 (1.8) | 70.81 |
| GPT-GNN [19] | 74.9 (0.3) | 62.5 (0.4) | 58.1 (0.3) | 58.3 (5.2) | 75.9 (2.3) | 65.2 (2.1) | 64.5 (1.4) | 77.9 (3.2) | 68.45 |
| ContextPred [18] | 73.9(0.5) | 62.8(0.3) | 59.9(1.6) | 74.3(3.2) | 72.4(1.8) | 75.6(1.0) | 70.8(1.4) | 78.5(1.3) | 71.03 |
| GraphLoG◇ [55] | 75.0(0.6) | 63.4(0.6) | 59.6(1.9) | 75.7(2.4) | 75.5(1.6) | 76.1(0.8) | 68.7(1.6) | 78.6(1.0) | 71.56 |
| G-Contextual [35] | 75.3(0.4) | 62.4(0.5) | 58.5(1.1) | 60.3(4.8) | 72.3(0.9) | 76.5(1.3) | 69.7(1.8) | 78.2(1.2) | 69.33 |
| G-Motif [35] | 73.2(0.6) | 62.0(0.8) | 61.1(1.2) | 77.5(2.5) | 73.4(1.6) | 73.3(1.5) | 66.6(2.6) | 73.3 (3.1) | 70.05 |
| AD-GCL [43] | 74.6(0.2) | 63.6(0.4) | 61.4(0.8) | 76.3 (2.4) | 72.4(1.5) | 75.8(1.0) | 69.5 (0.6) | 75.5(1.2) | 71.14 |
| KCL [11] | 74.5(0.3) | 62.7(0.7) | 59.6(0.9) | 65.5(5.5) | 73.4(2.6) | 75.7(0.6) | 65.0(1.1) | 74.0 (1.5) | 68.80 |
| GraphMAE◇ [17] | 75.2(0.9) | 63.6(0.3) | 60.5(1.2) | 76.5(3.0) | 76.4(2.0) | 76.8(0.6) | 71.2(1.0) | 78.2(1.5) | 72.30 |
| D-SLA◇ [23] | 75.3(0.4) | 63.2(0.3) | 60.8(1.2) | 76.6(2.8) | 76.2(1.5) | 76.6(1.4) | 69.8(0.8) | 78.3(1.4) | 72.10 |
| JOAO [57] | 74.8 (0.6) | 62.8 (0.7) | 60.4 (1.5) | 66.6 (3.1) | 76.6 (1.7) | 76.9 (0.7) | 66.4 (1.0) | 73.2 (1.6) | 69.71 |
| SimGRACE [51] | 74.4 (0.3) | 62.6 (0.7) | 60.2 (0.9) | 75.5 (2.0) | 75.4 (1.3) | 75.0 (0.6) | 71.0 (1.1) | 74.9 (2.0) | 71.15 |
| GraphCL [58] | 75.1 (0.7) | 63.0 (0.4) | 59.8 (1.3) | 77.5 (3.8) | 76.4 (0.4) | 75.1 (0.7) | 67.8 (2.4) | 74.6 (2.1) | 71.16 |
| MGSSL [61] | 75.2(0.6) | 63.3(0.5) | 61.6(1.0) | 77.1(4.5) | 77.6(0.4) | 75.8(0.4) | 68.8(0.6) | 78.8(0.9) | 72.28 |
| GraphMVP [28] | 75.9(0.5) | 63.1(0.2) | 60.2(1.1) | 79.1(2.8) | 77.7(0.6) | 76.0(0.1) | 70.8(0.5) | 79.3(1.5) | 72.76 |
| Vanilla Average | 73.8(1.0) | 60.2(0.7) | 58.5(1.3) | 57.0(5.2) | 71.5(0.9) | 75.2(1.7) | 65.6(1.1) | 70.9(1.8) | 66.59 |
| Concatenation | 75.5(0.7) | 62.7(1.0) | 62.8(0.9) | 77.8(3.5) | 76.3(0.6) | 75.7(1.3) | 70.3(0.7) | 77.9(1.1) | 72.38 |
| Ensemble | 76.1(0.1) | 64.3(0.2) | 63.1(1.0) | 78.2(1.5) | 77.8(0.2) | 77.1(0.3) | 71.4(0.5) | 77.6(0.8) | 73.20 |
| Model Fusion | 75.7(0.3) | 63.0(0.1) | 60.7(0.7) | 77.4(2.1) | 77.3(0.2) | 75.8(0.5) | 70.4(0.5) | 76.3(1.0) | 72.08 |
| **UniGM-F (RNN)** | 77.2+(0.4) | 64.9+(0.5) | **64.6**★(0.9) | **80.3**★(1.8) | 78.9+(1.1) | 77.6★ (0.8) | 71.3(0.5) | 80.4★(1.4) | 74.40 |
| **UniGM-T (RNN)** | **78.0**+(0.5) | **65.3**+(0.3) | 64.2★(1.3) | 79.5★(2.7) | **79.7**+(0.7) | **78.2**+(1.0) | **71.9**★(0.9) | **81.3**★(1.2) | **74.78** |

will over-fit the scarce samples; (2) The performance gains can be attributed to GMs' diversity because UniGM outperforms '4×MGSSL' by large margins.

**The number of GMs.** We also study the influence of the number of GMs by sequentially adding the following six GMs: EdgePred, InfoGraph, SimGRACE, GraphCL, GraphMVP, and MGSSL. We conduct experiments on Toxcast dataset. 'Best single model' refers to the GM whose performance is the best in the models' pool. As shown in Table 7, UniGM consistently outperforms the best single model. Additionally, UniGM performs better with more GMs. However, the memory consumption which increases with the number of GMs linearly will limit the practical applications.

**Alignment, aggregation module, and RAPNet.** For the alignment module of UniGM, we remove it and observe performance drops in Table 8. Additionally, we substitute RAPNet with an MLP-based policy network. Specifically, the MLP takes the output of the last layer as input and outputs the aggregation policy followed by softmax function. Also, we try various RNNs for RAPNet. RAPNet with RNNs outperforms MLP-based policy networks, verifying that modeling the dependency between different layers is necessary and conducive. Secondly, RAPNet with RNN performs better than LSTM and Gated Recurrent Unit (GRU) in general. For both UniGM and

**Table 2: Results for heterogeneous GMs. Model fusion and vanilla average cannot work in this setting.**

|  | Tox21 | ToxCast | Sider | ClinTox | MUV | HIV | BBBP | Bace | Average |
|---|---|---|---|---|---|---|---|---|---|
| GraphCL (6-layer GCN) | 74.2(0.6) | 61.5(0.7) | 61.3(1.7) | 75.0(3.6) | 76.3(0.9) | 74.6(0.7) | 65.6(2.1) | 71.2(3.9) | 70.01 |
| GraphMVP (3-layer GIN) | 72.6(0.4) | 60.2(0.4) | 58.3(1.1) | 63.6(3.6) | 72.1(1.1) | 74.2(0.6) | 64.1(1.5) | 65.7(2.2) | 66.35 |
| SimGRACE (5-layer GIN) | 74.4 (0.3) | 62.6 (0.7) | 60.2 (0.9) | 75.5 (2.0) | 75.4 (1.3) | 75.0 (0.6) | 71.0 (1.1) | 74.9 (2.0) | 71.15 |
| MGSSL (4-layer GraphSAGE) | 73.8(0.5) | 61.8(0.3) | 59.1(1.5) | 66.2(4.2) | 76.2(1.2) | 73.6(0.5) | 68.6(1.2) | 72.6(2.1) | 68.99 |
| Concatenation | 75.0(0.4) | 61.6(0.7) | 61.9(1.0) | 75.0(4.2) | 77.5(0.6) | 75.4(0.9) | 71.0(1.5) | 74.8(2.0) | 71.53 |
| Ensemble | 75.3(0.2) | 62.9(0.2) | 62.5(1.4) | 76.6(4.1) | 77.3(0.3) | 76.0(0.4) | 70.3(0.3) | 75.4(1.7) | 72.04 |
| **He-UniGM (5-layer GIN)** | 76.7$^{+}$(0.7) | 63.8$^{+}$(0.5) | 63.6$^{\star}$(0.7) | 75.4(2.5) | 78.5$^{+}$(1.2) | 77.6$^{+}$(0.8) | 71.6$^{+}$(1.2) | 77.5$^{\star}$(1.4) | 73.08 |

**Table 3: Results for protein function prediction and research field prediction.**

| Methods | No pre-train | Infomax | EdgePred | ContextPred | AttrMask | Concatenation | Model Fusion | Ensemble | **UniGM-F** | **UniGM-T** |
|---|---|---|---|---|---|---|---|---|---|---|
| **Protein function prediction** | 64.8(1.0) | 64.1(1.5) | 65.7(1.3) | 65.2(1.6) | 64.4(1.3) | 66.1(0.9) | 64.9(1.7) | 66.4(0.8) | 68.1(1.2) | **68.6**(1.4) |
| **Research field prediction** | 69.01(0.23) | 69.54(0.08) | 69.43(0.07) | 69.37 (0.21) | 68.61(0.16) | 69.91(0.25) | 68.14(0.09) | 70.21(0.11) | 71.69(0.20) | **72.85**(0.17) |

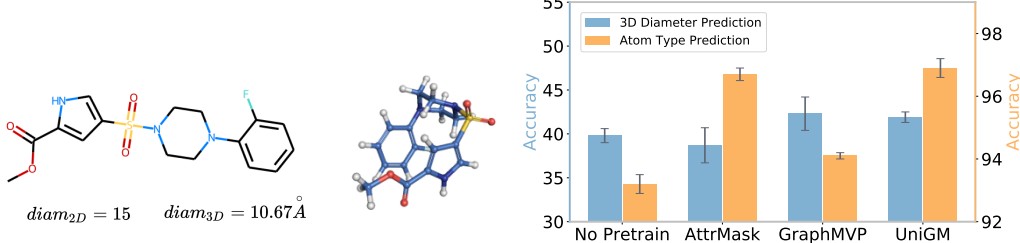

**Figure 4: Left: An example of 3D Diameter Prediction task in [28]. Right: The performance of GraphMVP, AttrMask and UniGM in the two tasks. UniGM acquire the specialized abilities of AttrMask and GraphMVP.**

**Table 4: Comparisons of training and inference time on the same device in the homogeneous setting.**

| Methods | **ToxCast** | | **Sider** | |
|---|---|---|---|---|
|  | Training | Inference | Training | Inference |
| Single GM | 368.3 s | 102.8 s | 88.1 s | 37.6 s |
| Model Fusion | 531.2 s | 115.5 s | 120.8 s | 39.9 s |
| Ensemble | 1536.7 s | 442.8 s | 370.1 s | 135.4 s |
| UniGM-T | 981.2 s | 211.7 s | 215.6 s | 64.8 s |
| UniGM-F | 778.4 s | 195.6 s | 176.5 s | 56.5 s |

**Table 5: Comparisons of training and inference time on the same device in the heterogeneous setting.**

| Methods | **ToxCast** | | **Sider** | |
|---|---|---|---|---|
|  | Training | Inference | Training | Inference |
| GraphCL (6-layer GCN) | 415.9 s | 116.8 s | 95.4 s | 46.3 s |
| GraphMVP (3-layer GIN) | 222.6 s | 64.5 s | 54.6 s | 25.8 s |
| SimGRACE (5-layer GIN) | 368.3 s | 102.8 s | 88.1 s | 37.6 s |
| MGSSL (4-layer GraphSAGE) | 235.7 s | 68.9 s | 61.8 s | 29.3 s |
| Ensemble | 1482.5 s | 361.2 s | 329.7 s | 151.8 s |
| He-UniGM | 916.6 s | 98.5 s | 205.7 s | 36.3 s |

**Table 6: The influence of GMs' diversity for UniGM.**

| Methods | 4 × MGSSL (UniGM-T) | 4 × MGSSL (UniGM-F) | UniGM-F | **UniGM-T** |
|---|---|---|---|---|
| Sider | 62.0(0.9) | 61.5(1.3) | **64.6(0.9)** | 64.2(1.3) |
| Toxcast | 63.1(0.8) | 63.0(0.1) | 64.9(0.5) | **65.3(0.3)** |
| Tox21 | 76.6(0.1) | 75.8(0.5) | 77.2(0.4) | **78.0(0.5)** |

He-UniGM, we replace the adaptive aggregation with vanilla average and random aggregation. The results indicate that the learned importance of each GM is meaningful.

## 4.5 Results for Pre-trained Models in Multiple Modalities

As we mentioned in the main text, our approaches are not limited to GNNs scenarios but could be flexibly applied to various scenarios

**Table 7: The influence of the number of GMs.**

| Num. of GMs | 2 | 3 | 4 | 5 | 6 |
|---|---|---|---|---|---|
| Best single model | 62.8(0.6) | 62.8(0.6) | 63.0(0.4) | 63.1(0.2) | 63.3(0.5) |
| **UniGM-F** | 63.3(0.5) | 64.2(0.2) | 64.5(0.3) | 64.0(0.6) | 64.5(1.1) |
| **UniGM-T** | 63.7(0.1) | 64.8(0.5) | 65.3(0.3) | 65.9(0.5) | 65.5(0.7) |

**Table 8: Ablations on alignment, RAPNet of UniGM, and the aggregation of He-UniGM.**

| Methods | Tox21 | Toxcast | Sider |
|---|---|---|---|
| UniGM-T w/o alignment | 75.4(1.0) | 63.8(0.1) | 61.5(2.1) |
| UniGM-T with MLP | 76.7(0.2) | 63.6(0.7) | 63.0(1.5) |
| UniGM-T with GRU | 77.2(0.2) | 64.5(0.5) | 62.9(0.7) |
| UniGM-T with LSTM | 77.7(0.6) | 64.8(0.3) | 63.8(1.0) |
| UniGM-T (Vanilla average) | 76.6(1.0) | 63.0(0.7) | 62.1(1.2) |
| UniGM-T (Random aggregation) | 76.4(0.8) | 63.5(0.9) | 62.6(0.6) |
| UniGM-T | **78.0**(0.5) | **65.3**(0.3) | **64.2**(1.3) |
| UniGM-F w/o alignment | 75.1(0.7) | 63.6(1.1) | 61.7(1.3) |
| UniGM-F with MLP | 75.9(0.7) | 63.4(0.9) | 63.5(0.9) |
| UniGM-F with GRU | 76.5(0.6) | 63.8(0.1) | 64.1(0.7) |
| UniGM-F with LSTM | **77.5**(0.8) | 64.3(0.4) | 64.3(1.1) |
| UniGM-F (Vanilla average) | 75.6(1.3) | 63.2(0.5) | 62.4(1.5) |
| UniGM-F (Random aggregation) | 75.4(1.1) | 63.0(0.7) | 62.0(1.6) |
| UniGM-F | 77.2 (0.4) | **64.9**(0.5) | **64.6**(0.9) |
| He-UniGM (Vanilla average) | 75.8(0.1) | 62.2(0.6) | 62.8(1.6) |
| He-UniGM (Random aggregation) | 75.4(0.7) | 62.5(0.3) | 62.5(1.1) |
| **He-UniGM** | **76.7**(0.7) | **63.8**(0.5) | **63.6**(0.7) |

**Table 9: UniGM for pre-trained vision models (top-1 accuracy).**

| Models | CIFAR-100 | COCO-70 |
|---|---|---|
| ImageNet Supervised | 81.18 | 81.97 |
| MOCO | 75.31 | 75.66 |
| Mask R-CNN | 79.12 | 81.64 |
| DeepLabV3 | 78.76 | 80.70 |
| Keypoint R-CNN | 76.38 | 76.53 |
| Model Fusion | 80.77 | 81.74 |
| Ensemble | 82.18 | 82.42 |
| **UniGM-F** | 83.56 | 83.86 |
| **UniGM-T** | **83.83** | **84.69** |

**Table 10: UniGM for pre-trained language models.**

| Models | SST-2 (Acc.) | RTE (Acc.) |
|---|---|---|
| BERT | 92.1 | 65.8 |
| RoBERTa | 92.9 | 68.9 |
| UniLM | 93.3 | 70.6 |
| Model Fusion | 93.5 | 71.9 |
| Ensemble | 93.8 | 72.7 |
| **UniGM-F** | 94.2 | 74.8 |
| **UniGM-T** | **94.6** | **75.7** |

such as in NLP or computer vision (CV). In this section, we unify the

pre-trained models in CV and NLP. For pre-trained vision models, we unify 5 representative pre-trained vision models: (1) Supervised pre-trained models on ImageNet [36]; (2) Unsupervised pre-trained models with MOCO [15] on ImageNet; (3) Mask R-CNN [16] model for detection and instance segmentation; (4) DeepLabV3 [5] model for semantic segmentation; (5) Keypoint R-CNN model for keypoint detection, pre-trained on COCO-2017 challenge datasets of each task. All these pre-trained models are from torchvision or original implementation. For pre-trained language models, we combine BERT [7], RoBERTa [29] and UniLM [8]. We conduct experiments on two text datasets with different sizes. The first one is SST-2 [39], which is a benchmark for text sentiment classification. The second one is RTE [3], which is a widely used dataset for natural language inference. The results can be seen in Table 9 and Table 10, from which we can observe that UniGM consistently outperforms each single model and competitive baselines including ensemble and model fusion. Compared with the graph domain, the superiority of UniGM in CV or NLP domains is even more pronounced.

## 4.6 Visualization Analysis

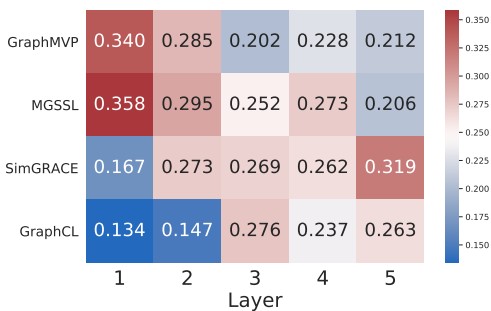

**Figure 5: Visualization of the learned aggregation policy.**

We visualize the learned aggregation policy for diverse GMs on Toxcast dataset in Figure 5. As can be observed, the policies vary significantly across different GMs and layers, which coincides with previous literatures that claim different pre-trained models have different relations to the downstream tasks and different layers can capture different knowledge [24, 60]. Concretely, GMs such as GraphMVP and MGSSL that introduce external knowledge outweigh the contrastive GMs including SimGRACE and GraphCL. Additionally, the higher layer of SimGRACE and GraphCL are generally more important for downstream tasks.

## 5 CONCLUSION

In this paper, we make one of the first attempts to unify multiple pre-trained GMs for better performance in downstream tasks. Specifically, we propose UniGM whose variants can integrate both homogeneous and heterogeneous pre-trained models into a unified one in an effective and efficient manner. The empirical results suggest that UniGM can achieve better performance in various downstream tasks. Currently, tremendous efforts are devoted to designing pre-training strategies for multiple modalities. Despite the fruitful progress, exploring more effective and efficient ways to leverage pre-trained models warrant further research in the future.

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
