# OpenReview forum: "UniGM: Unifying Multiple Pre-trained Graph Models via Adaptive Knowledge Aggregation"
_acmmm.org/ACMMM/2024/Conference — MM2024 Poster_

### Official Review · Reviewer_YoM4 · 2024-04-29

**Rating:** 4
**Confidence:** 3

**Summary:**

The paper presents UniGM, a novel approach for unifying multiple pre-trained Graph Models (GMs) to enhance performance in downstream tasks. UniGM introduces two techniques for integrating homogeneous and heterogeneous GMs, respectively, and demonstrates its effectiveness across various domains such as chemistry, biology, and bibliography. The empirical results show that UniGM consistently outperforms individual GMs and other aggregation methods like ensemble and model fusion.

**Strengths:**

1. UniGM provides a new perspective on leveraging diverse pre-trained GMs through adaptive knowledge aggregation.
2. The paper offers extensive experimental results, showcasing the superiority of UniGM over existing methods in multiple domains.
3. The paper is clearly organized, making it easy to follow the authors' line of reasoning and the results of their experiments.
4. The paper includes a thorough case study, ablation study, and visualization analysis, providing a deep understanding of the model's behavior.

**Limitations:**

1. The method presented in this paper is not explicitly tailored for graph learning, which the authors view as an advantage; however, I argue that this might not be a distinct strength. Furthermore, if there have been studies on similar problems in other fields, such as Natural Language Processing (NLP), necessary comparisons should be made with these methods.
2. The proposed method appears to be straightforward and intuitive. Are there any performance advantages compared to more complex counterparts, such as those utilizing the attention mechanism or its variants for computing importance?
3. Intuitively, the He-UniGM designed for heterogeneous scenarios can replace the UniGM and should explain the necessity of the UniGM.
4. The heterogeneous scenarios discussed in this paper are incomplete. Various pre-trained models differ not only in spatial resolution, channel number, and the number of model layers but also in patterns. To address these disparities, the authors may consult references such as Paper [1] for the spatial resolution, Paper [2] for the pattern gap and $1 \times 1$ convolution for the channel number.
5. Finally, the baselines used in the experiments are inadequate. To tackle the issues discussed in the paper, certain state-of-the-art feature fusion methods in Computer Vision may outperform the method proposed by the authors.

[1] Learning to resize images for computer vision tasks

[2] Cdtrans: Cross-domain transformer for unsupervised domain adaptation

**Suitability:**

2

---

### Official Review · Reviewer_W2UT · 2024-05-14

**Rating:** 5
**Confidence:** 4

**Summary:**

The paper introduces two methodologies for incorporating pre-trained models, with one specifically tailored to graph neural networks (GNNs) and the other being more general in nature, albeit with a focus on GNNs. This topic holds significance as pre-trained GNNs have demonstrated substantial advantages, yet this particular aspect is often overlooked. Notably, the observation regarding the prevalent use of similar base architectures across models is intriguing and effectively leveraged. Furthermore, the paper is commendably written, offering sufficient detail to warrant a recommendation for acceptance.

**Strengths:**

1. The previous researchers primarily focused on designing graph pre-training techniques. This article, for the first time, explores how to efficiently leverage multiple off-the-shelf graph pre-training models. The topic is important and the authors focus on an interesting aspect.

2.  The authors present a well-thought-out methodology that includes both homogeneous and heterogeneous GMs.

3. The extensive experiments across various domains demonstrate the superiority of UniGM over individual GMs and alternative ensemble approaches. The results are statistically significant.

4. The paper is well-written with a clear introduction, related work, methodological details, and a comprehensive experimental section.

5. The authors have made the code publicly available.

**Limitations:**

1. The paper could benefit from more discussion on related works regarding transfer learning.

2. Minor issue: The spacing between Line 81 and Line 83 is a bit too large.

3. The definition of "heterogeneous" in He-UniGM is confused. Could you provide clear defination for it?  In GNNs, a heterogeneous graph typically contains different types of entities (nodes) and relationships (edges) between these entities. What is your difference with the heterogeneous graph?

**Suitability:**

2

---

### Official Review · Reviewer_5yLn · 2024-05-23

**Rating:** 4
**Confidence:** 3

**Summary:**

The paper titled proposes a novel approach to leverage multiple pre-trained Graph Models (GMs) for improved performance in downstream tasks. The method involves two main strategies: unifying homogeneous GMs with the same architecture using a Recurrent Aggregation Policy Network (RAPNet), and unifying heterogeneous GMs with different architectures using an alignment module and a meta-learner. Extensive experiments demonstrate the effectiveness of UniGM in various domains, including chemistry, biology, and bibliography.

**Strengths:**

- The use of RAPNet for adaptive, sample-dependent, and layer-dependent aggregation is novel and addresses limitations of existing ensemble and model fusion methods.
- The paper provides extensive experimental results across multiple domains, showing consistent improvements over single models and competitive baselines.
- The proposed method is flexible and can be applied to various types of GMs, including those from different domains such as computer vision and natural language processing.

**Limitations:**

- How does the alignment module handle potential semantic differences in parameter matrices of the same layer from different pre-trained GMs? What criteria are used to ensure effective alignment?
- Can you provide more insights into how the learned aggregation weights are interpreted? Specifically, how do they vary across different layers and samples, and what does this variation indicate about the underlying model's learning process?
- How does the meta-learner determine the importance of each GM conditioned on the input sample? Are there specific features or characteristics of the input that the meta-learner prioritizes?
- When integrating GMs pre-trained on very different tasks or datasets, how does UniGM ensure that the knowledge from these diverse sources is effectively combined without conflicts?
- The authors should include additional related methods to further validate the effectiveness of the proposed approach: CLEAR: Cluster-enhanced Contrast for Self-supervised Graph Representation Learning. TNNLS'22; Graph Self-supervised Learning with Augmentation-aware Contrastive Learning. WWW'23
- As a research direction in machine learning and data mining on graphs, it is necessary to have some up-to-date surveys and related works. A Comprehensive Survey on Graph Neural Networks. 2024; A Comprehensive Survey on Deep Graph Representation Learning. 2023; Graph Neural Networks- Taxonomy, Advances and Trends. 2022; Deep Learning on Graphs: A Survey. 2022

**Suitability:**

2

---

### Official Review · Reviewer_TJqD · 2024-05-26

**Rating:** 4
**Confidence:** 3

**Summary:**

This paper presents a novel approach to integrating multiple Graph Models (GMs) in order to improve performance in downstream tasks. They introduce UniGM, a unified framework that can effectively and efficiently combine both homogeneous and heterogeneous pre-trained GMs.

**Strengths:**

This paper is clearly written and well-structured. It presents a novel approach to efficiently utilizing pre-trained Graph Models (GMs), which unify multiple GMs to enhance performance in downstream tasks. They propose two distinct and efficient techniques to unify both homogeneous and heterogeneous GMs. Extensive experiments confirm that UniGM consistently outperforms individual GMs with moderate computational consumption.

**Limitations:**

The paper's title might be confusing as it suggests a universal Graph Model capable of dealing with a variety of graph data and tasks. However, the work actually seems more like a distillation method with multiple Pre-trained Graph Models as teacher models.

The paper limits the application of the method to graph classification tasks only. While this task was explored in detail, the paper could have potentially broadened its scope to include other graph tasks, like node classification, where the method might also be applicable.

**Suitability:**

2

---

### Meta-Review · Area_Chair_QLJo · 2024-06-26

**Recommendation:** Accept (Poster)
**Confidence:** 4

**Metareview:**

This paper received four positive scores (ba,ba,wa,wa) from reviewers. This paper leverages multiple graph models to advance the performance in the downstream tasks. Reviewers believe that the paper is clearly written and well-structured. There are also some concerns from reviewers such as limited applications and missing related works. Authors should solve these issues in their final version.